# Internal Resonance of the Coupling Electromechanical Systems Based on Josephson Junction Effects

**DOI:** 10.3390/mi13111958

**Published:** 2022-11-11

**Authors:** Canchang Liu, Lijun Li, Yirui Zhang

**Affiliations:** School of Transportation and Vehicle Engineering, Shandong University of Technology, Zibo 255049, China

**Keywords:** coupling beam, Josephson junction, internal resonance

## Abstract

The internal resonances of the coupling vibration among electro-dynamic modes of an NEMS are studied for the coupling resonators connected on a Josephson junction. The methodology adopted involves coupling a resonator connected on a Josephson junction. The mathematical model of the coupled system is then obtained by considering the regulatory nonlinear effect of the phase difference of that Josephson junction. The resulting dynamic differential equation is deduced by considering the nonlinear terms of the Josephson junction and the nanobeam. The multi-scale method is then used to obtain the 1:1:1 resonant amplitude–frequency response equation of the coupled electromechanical system. The influence of the phase difference of the Josephson junction, magnetic field, external excitation and other factors are analyzed based on the internal resonant amplitude of the coupled system. The simulation results illustrate that the changes in the values of the magnetic field, excitation amplitude and divided resistances can lead to a remarkable change in the values of the nanobeam frequency and amplitude. The internal resonance principle is used to generate a mutual conversion and amplification among electrical signals and mechanical signals. This research provides a theoretical framework and a numerical approach for improving the sensitivity of magnetic quality detection.

## 1. Introduction

The superconducting Josephson junction effect and Josephson junction array devices have been widely used in the quantum voltage reference and the basic units of the international system of units in metrology. Moreover, superconducting microwave devices provide superior performance over traditional microwave devices, which have been widely used in mobile communication, radar and some special communication systems. By that means, the Josephson junction devices play an increasingly important role in the field of geophysics, astrophysics, quantum information technology, materials science, biomedicine and many other frontier fields [1]. The Josephson effects are widely used in the generation and reception of very high frequency signals, which are also a good candidate in the modeling and fabrication of very weak magnetic field sensors. This has been the subject of many areas of research recently [2,3,4,5,6].

The nonlinear resonance of a coupling structure is usually accompanied by the transfer and distribution of energy. Younis et al. studied the energy transfer phenomenon between resonator modes, which resulted in the establishment of the frequency stabilization of the micro electro-mechanical system (MEMS) resonator [7,8]. The coupling effect exists among different resonant modes in the vibration structure, and the driving energy can generate multi-steady energy transfer among these different resonant modes. The nonlinear mechanism plays an important role in the energy transfer process of the mechanical system [9], which is an important reason for the generation of modal energy concentration. Hu et al. pointed out that energy can be transferred between objects with internal resonances and can affect the nonlinear characteristics of the structure [10]. The study of the chaotic behavior in Duffing systems has been widely studied during the last few decades [11,12]. However, there are difficulties in pumping energy from the lower level to the higher level of stiffness for this class of electromechanical systems, mainly due to the mechanical structures representing very high values of stiffness and energy density. This requires a stronger work mode and a longer pumping time, which restricts its applications in the field of engineering.

The coupled micro-beam oscillators can produce resonance, and their theoretical and experimental studies have gradually attracted the attention of many scientific researchers [13,14,15,16,17]. Some research works aim firstly to analyze the resonance conditions resulting from these types of oscillators along with their characteristics. Huan et al. pointed out that the phase delay, the nonlinear coefficient and the excitation amplitude can effectively regulate the width and the position of the resonance interval of the nonlinear oscillator. Moreover, a temperature compensation circuit was already designed to fine-tune the synchronous resonance working interval of the resonator by using the Joule heat effect [13]. Along similar lines, the double-mechanically coupled cantilever beam can generate a nonlinear resonance frequency multiplication effect with a low frequency excitation and a high frequency output [14]. The superharmonic vibration of a single beam has high amplitudes at the output, which is conducive to the improvement of the sensor sensitivity and resolution [14,16]. Wei et al. studied the triggering conditions of the internal resonance frequency locking, the amplitude range of that locking frequency and the influence of coupling and excitation intensity on the amplitude range [17]. Other research works aim as well to study the synchronous resonance of a coupled microbeam. Wei et al. also studied the 1/121 order synchronous vibration [17], whereas Shim et al. studied the 1/7 order synchronous vibration [18]. The results illustrate that the resonance interval decreases with the increase in the frequency ratio. Note that, for higher order beam synchronizations, resonances are more difficult to perform. Indeed, the above studies found that the frequency band range of a nonlinear resonance is narrow. The narrower frequency band of the highest order resonance brings challenges in terms of controlling the synchronous tuning of the coupling structure in question.

The coupling resonance of MEMS has become one of the hot topics in research. The nonlinear coupling and energy transfers between multiple modes in micro/nano-mechanical resonators were studied on intermodal coupling, internal resonance and synchronization [19,20,21,22,23]. Multimode nonlinear coupling was achieved by (1:2) internal resonance and parametric excitation with efficient coherent energy transfer [20]. The influence of residual stress on the modal characteristics of MEMS resonator array was analyzed with the laser Doppler method [21]. Asadi et al. investigated experimentally and analytically the 1:2 and 2:1 internal resonance in a clamped–clamped beam resonator to provide insights into the detailed mechanism of internal resonance [22]. A theoretical model was developed to depict the scale effect on the intermodal coupling in nanomechanical resonators based on the nonlocal theory of elasticity [23]. A size-dependent electrostatic model for cantilever micro-actuated beams was investigated considering the microstructure and surface energy effects [24]. The generalized differential quadrature method was employed to investigate the static and dynamic pull-in instability of nano-switches with small width/height ratios [25]. The strain gradient continuum theory was employed to investigate the size dependent pull-in instability of beam-type nano-electromechanical systems [26].

The Josephson junction is a suitable choice among many other mechanical devices. This has been the subject of intensive research recently [27,28,29,30,31]. Moreover, there are multiple applications for Josephson junctions in various fields such as engineering, health and communication. Koudafokê et al. studied the influence of a Josephson junction on the dynamic modes of the micro-beam oscillator [27]. They also modeled and analyzed the electro dynamic modes of a self-sustaining active sensor with Josephson junctions [28]. Zhang et al. proposed a Josephson junction-based feasible neuron to estimate the effect of the magnetic field. Along similar lines, a magnetic flux-controlled memristor connected in parallel with an ideal Josephson junction was used to percept the induction currents induced by the magnetic field [2,3]. The Josephson effects are widely used for the generation and the reception of very high frequency signals. In addition, the dynamics of three and four non-identical Josephson junctions connected in series and coupled with an RLC dipole were investigated [30]. Yamapi et al. studied the noise effect of a biorhythmic Josephson junction coupled to a resonator. They have found that the stability analysis of the Josephson junction coupled to a resonator shows a striking change in the biorhythmic region. The coupled oscillators with nonlinearity of sinus similar to Josephson junctions were also discussed in the context of designing chaotic generators in [32].

The measurement of magnetic fields has become a hot topic in the field of engineering technology. The external magnetic field can affect the performances of electromechanical nano-bridges sensors and switches [33]. The dynamic electromagnetic instability of nano-sensors immersed in an external magnetic flux was simulated [34]. The fundamental equations of motion of a doubly clamped CNT-based nano-sensor were calculated with regard to the nonlocal elasticity by using Hamilton’s principle and the Euler–Bernoulli beam model [35].

Inspired by this rich literature, the aim of this paper is then proposed to model and generate the dynamic modes of an active sensor whose frequencies satisfy the internal resonance conditions and whose oscillation frequencies are controlled by a Josephson junction. The aim of this work is then to study the nonlinear dynamical behavior of the nanobeam coupled to a Josephson junction, which can show the possibilities of control of the frequencies and amplitudes of vibrations of the nanobeam in order to make more dynamic and effective these nanomachines in their fields of use. The dynamic and electrical oscillation modes of a NEMS are obtained by coupling a Josephson junction with a resonator. By that means, the study of the internal resonances of the coupling electromechanical vibration is conducted in order to find the influence relations on the phase difference of the Josephson junction along with the magnetic field, the external excitation and other factors on the nonlinear behavior of the coupled system. This paper provides a magnetic field strength detection method, which provides a high sensitivity and a fast on-line test method. This methodological approach can be applied to magnetic field detection in medicine, geology, mining and mechanical fields, which is beneficial for the improvement of sensitivity detection.

## 2. Model of Coupling System

The NEMS (Nano Electro-Mechanical System) that we propose to study is presented in Figure 1. This NEMS is obtained by coupling resonators at a Josephson junction [1]. The set of two resonators mounted in series and coupled to a Josephson junction consist of a mechanical resonator and an electric resonator. All the parameters related to the nano-beam will be calculated based on the formulas well enumerated in [1,36].

According to Kirchhoff’s laws and the differential equation of the dynamics of the nanobeams (Euler Bernoulli model), we have [1,37]
(1)$$I_{0}\cos\omega t=i_{j}+i_{c}$$
(2)$$u_{g}=u_{p}+u_{L_{0}}+u_{C_{0}}$$
where, *I*_0_ is amplitude of the excitation current. *ω* is the frequency of the excitation. *t* is time. *i*_j_ is instantaneous current through the Josephson junction. *i*_c_ is instantaneous current through electrical and mechanical resonators. *u*_g_, *u*_p_, $u_{L_{0}}$ and $u_{C_{0}}$ are the instantaneous electrical voltage across the generator, instantaneous electrical voltage across the beam, instantaneous electrical voltage across *C*_0_ and *L*_0_, respectively.

Considering Equations (1) and (2), phase difference of the Josephson junction and coupled circuit equations can be written as
(3)$$\ddot{\bar{\varphi }}(t)+\frac{1}{R_{j}C_{j}}\dot{\bar{\varphi }}(t)+\frac{2eI_{C_{j}}}{\hbar C_{j}}\sin\bar{\varphi }+\frac{2e}{\hbar C_{j}}\dot{\bar{q}}-\frac{4eBA}{\hbar C_{j}\mu L}\;\int_{0}^{L}\frac{\partial w(z,t)}{\partial t}dz=\frac{2eI_{0}}{\hbar C_{j}}\cos(\omega t)$$
(4)$$\ddot{\bar{q}}+\frac{\left(r_{0}+r_{p}\right)}{L_{0}}\dot{\bar{q}}+\frac{1}{L_{0}C_{0}}\bar{q}-\frac{2BA}{\mu L}\int_{0}^{L}\frac{\partial^{2}w}{\partial t^{2}}dz-\frac{\hbar }{2eL_{0}}\dot{\bar{\varphi }}-\frac{2BAr_{0}}{\mu LL_{0}}\int_{0}^{L}\frac{\partial w}{\partial t}dz=0$$
where, $\dot{(})=\partial /\partial t$, $\ddot{(})=\partial^{2}/\partial t^{2}$. $\bar{\varphi }$ is phase difference of the Josephson junction (PDJJ). $\bar{q}$ is the circuit power. *R*_j_ is resistance of the Josephson junction. *B* is the magnetic field vector. *C*_j_ is self-capacitance of the Josephson junction. $I_{C_{j}}$ is critical current of the Josephson junction. *e* is elementary electric charge. *C*_0_ is capacitance. *μ* is resistivity of the micro-beam. *z* is vertical coordinate of the nanobeam. *r*_0_ is electrical resistance of the divided resistor. *r*_p_ is electrical resistance of the micro-beam. *L*, *d* and *h* are length wide and high of the micro-beam. *ħ* is Planck constant. *w* is the deflection of the beam. *L*_0_ is coil inductance. *A* is the cross sectional area of the nanobeam.

Considering the geometric nonlinearity of nanobeam, the dynamic differential equation of nanobeam can be written as
(5)$$EI_{y}\frac{\partial^{4}w}{\partial z^{4}}+\rho A\frac{\partial^{2}w}{\partial t^{2}}+\lambda \frac{\partial w}{\partial t}-\frac{EA}{2L}\int_{0}^{L}(\frac{\partial w}{\partial z}{)}^{2}dx\frac{\partial^{2}w}{\partial z^{2}}=F(t)$$
where, *E*, *ρ* and *λ* are Young module, volume density, resistivity and damping coefficient of the micro-beam. *I*_y_ is the moment of inertia of the nanobeam. *F* is the excitation for the nanobeam.

Both the ends of nanobeam are fixed-fixed boundary, which are w(0,t)=w(L,t)=0,
$w^{\prime }(0,t)=w^{\prime }(L,t)=0$. The deflection of beam *w*(*z*, *t*) can be written as
(6)$$w(z,t)=\sum_{n=1}^{\infty }Z_{n}(z){\bar{u}}_{n}(t)$$
where, *n* is the vibration mode. ${\bar{u}}_{n}(t)$ is the generalized coordinate of the amplitude. *Z_n_*(*z*) is the set of eigenfunctions of the equation.

The deflection of the nanobeam can be written as
(7)$$Z_{n}=\sqrt{\frac{2}{3}}[1-\cos(\frac{2\pi z}{L})]$$

Taking the phase difference of the Josephson junction as a small quantity, the sine expansion of this variable is expanded as a Taylor series. Substituting the deflection of the nanobeam into Equations (3)–(5), the modal orthogonalization transformation can be obtained
(8)$$\ddot{\bar{\varphi }}+\alpha_{1}\dot{\bar{\varphi }}+\omega_{\varphi }^{2}\bar{\varphi }-\frac{\omega_{\varphi }^{2}}{6}{\bar{\varphi }}^{3}+\alpha_{2}\dot{\bar{q}}-\alpha_{3}\dot{u}=f_{0}\cos(\omega t),$$
(9)$$\ddot{\bar{q}}+(\beta_{1}-\gamma_{3}\beta_{3})\dot{\bar{q}}+\omega_{q}^{2}\bar{q}-\beta_{2}\dot{\bar{\varphi }}+(\gamma_{1}\beta_{3}-\beta_{4})\dot{u}+\gamma_{2}\beta_{3}u=0$$
(10)$$\ddot{\bar{u}}+\gamma_{1}\dot{\bar{u}}+\omega_{u}^{2}\bar{u}-\gamma_{3}\dot{\bar{q}}+\gamma_{4}{\bar{u}}^{3}=0$$
where,$\alpha_{1}=\frac{1}{R_{j}C_{j}}$, $\omega_{\varphi }^{2}=\frac{2eI_{C_{j}}}{\hbar C_{j}}$, $\alpha_{2}=\frac{2e}{\hbar C_{j}}$, $\alpha_{3}=\sqrt{\frac{2}{3}}\frac{4eBA}{\hbar C_{j}\mu }\;$, $f_{0}=\frac{2eI_{0}}{\hbar C_{j}}$, $\beta_{1}=\frac{\left(r_{0}+r_{p}\right)}{L_{0}}$, $\omega_{q}^{2}=\frac{1}{L_{0}C_{0}}$, $\beta_{2}=\frac{\hbar }{2eL_{0}}$, $\beta_{3}=\sqrt{\frac{2}{3}}\frac{2BA}{\mu }$, $\beta_{4}=\sqrt{\frac{2}{3}}\frac{2BAr_{0}}{\mu L_{0}}$, $\gamma_{1}=\left(\frac{\lambda }{\rho A}+\frac{4B^{2}L}{3\mu \rho }\right)$, $\omega_{u}^{2}=\sqrt{\frac{16EI\pi^{4}}{3\rho AL^{4}}}$, $\gamma_{3}=\sqrt{\frac{3}{2}}\frac{BL}{\rho A}$, $\gamma_{4}=\frac{8E\pi^{4}}{9\rho L^{4}}$.

The dimensionless form of the above equation is given by $\tau =\omega_{0}t$, $\varphi =\frac{\bar{\varphi }}{\varphi_{0}}$, $U=\frac{\bar{u}}{u_{0}}$ and $Q=\frac{\bar{q}}{Q_{0}}$. We get
(11)$$\ddot{\varphi }+j_{1}\dot{\varphi }+\omega_{1}^{2}\varphi -j_{2}\varphi^{3}+j_{3}\dot{Q}-j_{4}\dot{U}=f\cos(\Omega t)$$
(12)$$\ddot{Q}+k_{1}\dot{Q}+\omega_{2}^{2}Q-k_{3}\dot{\varphi }+k_{4}\dot{U}+k_{5}U=0$$
(13)$$\ddot{U}+l_{1}\dot{U}+\omega_{3}^{2}U-l_{3}\dot{Q}+l_{4}U^{3}=0$$
where,$j_{1}=\frac{\alpha_{1}}{\omega_{0}}$, $\omega_{1}^{2}=\frac{\omega_{\varphi }^{2}}{\omega_{0}^{2}}$, $j_{2}=\frac{\omega_{\varphi }^{2}\varphi_{0}^{2}}{6\omega_{0}^{2}}$
$j_{3}=\frac{\alpha_{2}Q_{0}}{\omega_{0}\varphi_{0}}$, $j_{4}=\frac{\alpha_{3}u_{0}}{\omega_{0}\varphi_{0}}$, $f=\frac{f_{0}}{\omega_{0}^{2}\varphi_{0}}$, $\Omega =\frac{\omega }{\omega_{0}}$, $k_{1}=\frac{\left(\beta_{1}-\gamma_{3}\beta_{3}\right)}{\omega_{0}}$, $\omega_{2}={\left(\frac{\omega_{q}}{\omega_{0}}\right)}^{2}$, $k_{3}=\frac{\beta_{2}\varphi_{0}}{\omega_{0}Q_{0}}$, $k_{4}=\frac{(\gamma_{1}\beta_{3}-\beta_{4})u_{0}}{\omega_{0}Q_{0}}$, $k_{5}=\frac{\omega_{u}^{2}\beta_{3}u_{0}}{\omega_{0}^{2}Q_{0}}$, $\omega_{3}=\frac{\omega_{u}^{2}}{\omega_{0}^{2}}$, $l_{1}=\frac{\gamma_{1}}{\omega_{0}}$, $l_{3}=\frac{\gamma_{3}Q_{0}}{\omega_{0}u_{0}}$, $l_{4}=\frac{\gamma_{4}u_{0}^{2}}{\omega_{0}^{2}}$.

Letting $\varphi =v-\varphi_{e}$, $Q=q-Q_{e}$, $U=u-U_{e}$ and introducing a small parameter, the dynamics equations near the equilibrium point can be written as
(14)$$\ddot{v}+\omega_{1}^{2}v+\varepsilon (j_{1}\dot{v}-j_{2}v^{3}+j_{3}\dot{q}-j_{4}\dot{u}+j_{5}v^{2}+j_{6}v+j_{7})=\varepsilon f\cos(\Omega t)$$
(15)$$\ddot{q}+\omega_{2}^{2}q+\varepsilon (k_{1}\dot{q}-k_{3}\dot{\varphi }+k_{4}\dot{u}+k_{5}u-k_{6})=0$$
(16)$$\ddot{u}+\omega_{3}^{2}u+\varepsilon (l_{1}\dot{u}-l_{3}\dot{q}+l_{4}u^{3}-l_{5}u^{2}+l_{6}u-l_{7})=0$$
where, $j_{5}=3j_{2}\varphi_{e}$, $j_{6}=-3j_{2}\varphi_{e}^{2}$, $j_{7}=j_{2}\varphi_{e}^{3}-\omega_{1}^{2}\varphi_{e}$, $k_{6}=\omega_{2}^{2}Q_{e}+k_{5}U_{e}$, $l_{5}=3l_{4}U_{e}$, $l_{6}=3l_{4}U_{e}^{2}$, $l_{7}=\omega_{3}^{2}U_{e}+l_{4}U_{e}^{3}$. ε is a small parameter.

## 3. Internal Resonance of Coupling Electromechanical System

The approximate solutions of Equations (13)–(15) are expressed in the following forms
(17)$$v\left(t,\varepsilon \right)=v_{0}\left(T_{0},T_{1}\right)+\varepsilon v_{1}\left(T_{0},T_{1}\right)+\cdot \cdot \cdot$$
(18)$$q\left(t,\varepsilon \right)=q_{0}\left(T_{0},T_{1}\right)+\varepsilon q_{1}\left(T_{0},T_{1}\right)+\cdot \cdot \cdot$$
(19)$$u\left(t,\varepsilon \right)=u_{0}\left(T_{0},T_{1}\right)+\varepsilon u_{1}\left(T_{0},T_{1}\right)+\cdot \cdot \cdot$$
where, ε is a small parameter, $T_{0}=t$ represents a fast changing time scale. $T_{1}=\varepsilon t$ represents a slow changing time scale.

The frequency of the adjusting external excitation and the natural frequency of the beam form a primary resonance relationship. The natural frequency of the beam and the natural frequency of the circuit system form a 1:1:1 internal resonance relationship [28]. The internal resonance frequencies satisfy the following forms
(20)$$\Omega =\omega_{1}+\varepsilon \sigma_{1},\;\omega_{2}=\omega_{1}+\varepsilon \sigma_{2},\;\omega_{3}=\omega_{1}+\varepsilon \sigma_{3}$$
where, *σ*_1_, *σ*_2_ and *σ*_3_ are excitation frequency tuning parameters.

Substituting Equation (14) and its derivative with respect to time into Equation (15) and equalizing the same power of the left and right sides of Equations (14)–(16) lead to
(21)$$O\left(\varepsilon^{0}\right):D_{0}^{2}v_{0}+\omega_{1}^{2}v_{0}=0$$
(22)$$O\left(\varepsilon^{0}\right):D_{0}^{2}q_{0}+\omega_{2}^{2}q_{0}=0$$
(23)$$O\left(\varepsilon^{0}\right):D_{0}^{2}u_{0}+\omega_{3}^{2}u_{0}=0$$
(24)$$\begin{array}{l} O\left(\varepsilon^{1}\right):D_{0}^{2}v_{1}+\omega_{1}^{2}v_{1}=-2D_{0}D_{1}v_{0}-j_{1}D_{0}v_{0}+j_{2}v_{0}^{3}\\ -j_{3}D_{0}q_{0}+j_{4}D_{0}u_{0}-j_{5}v_{0}^{2}-j_{6}v_{0}-j_{7}+f\cos\Omega t \end{array}$$
(25)$$\begin{array}{l} O\left(\varepsilon^{1}\right):D_{0}^{2}q_{1}+\omega_{2}^{2}q_{1}=-2D_{0}D_{1}q_{0}\\ -k_{1}D_{0}q_{0}+k_{3}D_{0}\varphi_{0}-k_{4}D_{0}u_{0}-k_{5}u_{0}+k_{6} \end{array}$$
(26)$$\begin{array}{l} O\left(\varepsilon^{1}\right):D_{0}^{2}u_{1}+\omega_{3}^{2}u_{1}=-2D_{0}D_{1}u_{0}\\ -l_{1}D_{0}u_{0}+l_{3}D_{0}q_{0}-l_{4}u_{0}^{3}+l_{5}u_{0}^{2}-l_{6}u_{0}+l_{7} \end{array}$$

The approximate solution of Equation (21) is expressed as follows
(27)$$v_{0}=A_{1}\left(T_{1}\right)e^{i\omega_{1}T_{0}}+{\bar{A}}_{1}\left(T_{1}\right)e^{-i\omega_{1}T_{0}}$$

The approximate solution of Equation (22) is expressed as follows
(28)$$q_{0}=A_{2}\left(T_{1}\right)e^{i\omega_{2}T_{0}}+{\bar{A}}_{2}\left(T_{1}\right)e^{-i\omega_{2}T_{0}}$$

The approximate solution of Equation (23) is expressed as follows
(29)$$u_{0}=A_{3}\left(T_{1}\right)e^{i\omega_{3}T_{0}}+{\bar{A}}_{3}\left(T_{1}\right)e^{-i\omega_{3}T_{0}}$$

In the formula, ${\bar{A}}_{i}$ is the complex conjugate of $A_{i}$ (*i* = 1, 2, 3), and the polar coordinate form $A_{i}$ is
(30)$$A_{1}=\frac{1}{2}a_{1}e^{i\theta_{1}},\;A_{2}=\frac{1}{2}a_{2}e^{i\theta_{2}},\;A_{3}=\frac{1}{2}a_{3}e^{i\theta_{3}}$$

In order to avoid the occurrence of the secular term, substituting Equations (27)–(30) into Equations (19)–(21) yields the following forms
(31)$$\begin{array}{l} -2i\omega_{1}D_{1}A_{1}-j_{1}i\omega_{1}A_{1}+3j_{2}A_{1}^{2}{\bar{A}}_{1}-j_{3}i\omega_{2}A_{2}e^{j\sigma_{2}T_{0}}\\ +j_{4}i\omega_{3}A_{3}e^{j\sigma_{3}T_{0}}-j_{6}A_{1}+\frac{1}{2}fe^{j\sigma_{1}T_{1}}=0 \end{array}$$
(32)$$\begin{array}{l} -2i\omega_{2}D_{1}A_{2}e^{i\omega_{2}T_{0}}-k_{1}i\omega_{2}A_{2}e^{i\omega_{2}T_{0}}+k_{3}i\omega_{1}A_{1}e^{i\omega_{1}T_{0}}\\ -k_{4}i\omega_{3}A_{3}e^{i\omega_{3}T_{0}}-k_{5}A_{3}e^{i\omega_{3}T_{0}}=0 \end{array}$$
(33)$$\begin{array}{l} -2i\omega_{3}D_{1}A_{3}e^{i\omega_{3}T_{0}}-l_{1}i\omega_{3}A_{3}e^{i\omega_{3}T_{0}}+l_{3}i\omega_{2}A_{2}e^{i\omega_{2}T_{0}}\\ -3l_{4}A_{3}^{2}{\bar{A}}_{3}e^{i\omega_{3}T_{0}}-l_{6}A_{3}e^{i\omega_{3}T_{0}}=0 \end{array}$$

Substituting Equations (27) and (30) into Equation (31) and separating the real and imaginary parts yield into
(34)$$D_{1}a_{1}=-\frac{j_{1}}{2}a_{1}-\frac{j_{3}\omega_{2}}{2\omega_{1}}a_{2}\cos\phi_{1}+\frac{j_{4}\omega_{3}}{2\omega_{1}}a_{3}\cos\phi_{2}+\frac{f}{2\omega_{1}}\sin\phi$$
(35)$$a_{1}D_{1}\phi =(\sigma_{1}+\frac{j_{6}}{2\omega_{1}})a_{1}-\frac{3j_{2}}{8\omega_{1}}a_{1}^{3}-\frac{j_{3}\omega_{2}}{2\omega_{1}}a_{2}\sin\phi_{1}+\frac{j_{4}\omega_{3}}{2\omega_{1}}a_{3}\sin\phi_{2}-\frac{f}{2\omega_{1}}\cos\phi$$
where, $\phi_{1}=\theta_{2}-\theta_{1}+\sigma_{2}T_{1}$, $\phi_{2}=\theta_{3}-\theta_{1}+\sigma_{3}T_{1},\phi =\sigma_{1}T_{1}-\theta_{1}$.
(36)$$-\frac{j_{1}}{2}a_{1}-\frac{j_{3}\omega_{2}}{2\omega_{1}}a_{2}\cos\phi_{1}+\frac{j_{4}\omega_{3}}{2\omega_{1}}a_{3}\cos\phi_{2}+\frac{f}{2\omega_{1}}\sin\phi =0$$
(37)$$(\sigma_{1}+\frac{j_{6}}{2\omega_{1}})a_{1}-\frac{3j_{2}}{8\omega_{1}}a_{1}^{3}-\frac{j_{3}\omega_{2}}{2\omega_{1}}a_{2}\sin\phi_{1}+\frac{j_{4}\omega_{3}}{2\omega_{1}}a_{3}\sin\phi_{2}-\frac{f}{2\omega_{1}}\cos\phi =0$$

Eliminating the phase *φ*, the amplitude–frequency equation of the phase difference variable of the superconductor is
(38)$$\begin{array}{l} {\left(\frac{j_{1}}{2}a_{1}+\frac{j_{3}\omega_{2}a_{2}\cos\phi_{1}-j_{4}\omega_{3}a_{3}\cos\phi_{2}}{2\omega_{1}}\right)}^{2}+[(\sigma_{1}+\frac{j_{6}}{2\omega_{1}})a_{1}\\ -\frac{3j_{2}}{8\omega_{1}}a_{1}^{3}-\frac{j_{3}\omega_{2}a_{2}\sin\phi_{1}-j_{4}\omega_{3}a_{3}\sin\phi_{2}}{2\omega_{1}}{]}^{2}={\left(\frac{f}{2\omega_{1}}\right)}^{2} \end{array}$$

For the electric resonator system, as the electromotive force generated by the nanobeam is weak, and the effect of the magnetic field term can be ignored, so it can be obtained as
(39)$${\left(\frac{j_{1}}{2}a_{1}+\frac{j_{3}\omega_{2}a_{2}\cos\phi_{1}}{2\omega_{1}}\right)}^{2}+{\left[(\sigma_{1}+\frac{j_{6}}{2\omega_{1}})a_{1}-\frac{3j_{2}}{8\omega_{1}}a_{1}^{3}-\frac{j_{3}\omega_{2}a_{2}\sin\phi_{1}}{2\omega_{1}}\right]}^{2}={\left(\frac{f}{2\omega_{1}}\right)}^{2}$$

By substituting Equations (28) and (30) into Equation (32), the real and imaginary parts are separated into
(40)$$D_{1}a_{2}=-\frac{k_{1}}{2}a_{2}+\frac{k_{3}\omega_{1}}{2\omega_{2}}a_{1}\cos\phi_{3}-\frac{k_{4}\omega_{3}+k_{5}}{2\omega_{2}}a_{3}\cos\phi_{4}$$
(41)$$a_{2}D_{1}\phi_{4}=-\sigma_{23}a_{2}+\frac{k_{3}\omega_{1}}{2\omega_{2}}a_{1}\sin\phi_{3}-\frac{k_{4}\omega_{3}+k_{5}}{2\omega_{2}}a_{3}\sin\phi_{4}$$
where, $\phi_{3}=(\sigma_{1}-\sigma_{2})T_{1}-\theta_{2},\;\phi_{4}=(\sigma_{3}-\sigma_{2})T_{1}-\theta_{2}$, $\sigma_{23}=(\sigma_{2}-\sigma_{3})$. Let $\phi_{3}=\phi_{4}$, we get
(42)$$D_{1}a_{2}=-\frac{k_{1}}{2}a_{2}+\frac{k_{3}\omega_{1}a_{1}+(k_{4}\omega_{3}+k_{5})a_{3}}{2\omega_{2}}\cos\phi_{3}$$
(43)$$a_{2}D_{1}\phi_{4}=\sigma_{23}a_{2}+\frac{k_{3}\omega_{1}a_{1}+(k_{4}\omega_{3}+k_{5})a_{3}}{2\omega_{2}}\sin\phi_{3}$$

By canceling the phase *φ*_3_, the amplitude–frequency equation of the phase difference variable of the superconductor is
(44)$${\left(\frac{k_{1}}{2}\right)}^{2}a_{2}^{2}+\sigma_{23}^{2}a_{2}^{2}={\left[\frac{k_{3}\omega_{1}a_{1}-(k_{4}\omega_{3}+k_{5})a_{3}}{2\omega_{2}}\right]}^{2}$$

As the effect of the magnetic field term is ignored, it can be obtained as
(45)$${\left(\frac{k_{1}}{2}\right)}^{2}a_{2}^{2}+\sigma_{23}^{2}a_{2}^{2}={\left[\frac{k_{3}\omega_{1}a_{1}}{2\omega_{2}}\right]}^{2}$$

Let the tuning parameters $\sigma_{3}-\sigma_{2}=0$, and the peak amplitude can be obtained as
(46)$$a_{2\max}=\frac{k_{3}\omega_{1}a_{1}}{k_{1}\omega_{2}}$$

Equation (39) can be simplified as
(47)$$\begin{array}{l} {\left(\frac{j_{1}}{2}a_{1}+\frac{j_{3}k_{3}\cos\phi_{1}}{2k_{1}}a_{1}\right)}^{2}+[(\sigma_{1}+\frac{j_{6}}{2\omega_{1}})a_{1}-\\ \frac{3j_{2}}{8\omega_{1}}a_{1}^{3}-\frac{j_{3}k_{3}\sin\phi_{1}}{2k_{1}}a_{1}{]}^{2}={\left(\frac{f}{2\omega_{1}}\right)}^{2} \end{array}$$

By substituting Equations (29) and (30) into Equation (33), the real and imaginary parts are separated into
(48)$$D_{1}a_{3}=-\frac{l_{1}}{2}a_{3}+\frac{l_{3}\omega_{2}}{2\omega_{3}}a_{2}\cos\phi_{5}$$
(49)$$a_{3}D_{1}\phi_{5}=\sigma_{23}a_{3}+\frac{l_{6}}{2\omega_{3}}a_{3}+\frac{3l_{4}}{8\omega_{3}}a_{3}^{3}-\frac{l_{3}\omega_{2}}{2\omega_{3}}a_{2}\cos\phi_{5}$$
where, $\phi_{5}=(\sigma_{2}-\sigma_{3})T_{1}-\theta_{3}$.
(50)$$D_{1}a_{3}=-\frac{l_{1}}{2}a_{3}+\frac{l_{3}\omega_{2}}{2\omega_{3}}a_{2}\cos\phi_{5}$$
(51)$$a_{3}D_{1}\phi_{5}=\sigma_{23}a_{3}+\frac{l_{6}}{2\omega_{3}}a_{3}+\frac{3l_{4}}{8\omega_{3}}a_{3}^{3}-\frac{l_{3}\omega_{2}}{2\omega_{3}}a_{2}\cos\phi_{5}$$

Eliminating the phase *φ*_5_, the amplitude–frequency equation of the phase difference variable of the superconductor is
(52)$${\left(\frac{l_{1}}{2}\right)}^{2}a_{3}^{2}+{\left[\sigma_{23}+\frac{l_{6}}{2\omega_{3}}+\frac{3l_{4}}{8\omega_{3}}a_{3}^{2}\right]}^{2}a_{3}^{2}={\left(\frac{l_{3}\omega_{2}}{2\omega_{3}}a_{2}\right)}^{2}$$

Equation (52) can be simplified as
(53)$${\left(\frac{l_{1}}{2}\right)}^{2}a_{3}^{2}+{\left[\sigma_{23}+\frac{l_{6}}{2\omega_{3}}+\frac{3l_{4}}{8\omega_{3}}a_{3}^{2}\right]}^{2}a_{3}^{2}={\left(\frac{l_{3}k_{3}\omega_{1}}{2k_{1}\omega_{3}}a_{1}\right)}^{2}$$

The vibration of the nanobeam is driven by changing the Jefferson excitation signal to produce a changing circuit signal.

## 4. Equilibrium Points

Equations (14)–(16) for the non-autonomous system are written as [1]
(54)$${\dot{x}}_{1}=\dot{v}=x_{2}$$
(55)$${\dot{x}}_{2}=-j_{1}x_{2}-\omega_{1}^{2}x_{1}+j_{2}x_{1}^{3}-j_{3}y_{2}+j_{4}z_{2}-j_{5}x_{1}^{2}-j_{6}x_{1}-j_{7}+f\cos(\Omega t)$$
(56)$${\dot{y}}_{1}=\dot{q}=y_{2}$$
(57)$${\dot{y}}_{2}=-k_{1}y_{2}-\omega_{2}^{2}y_{1}+k_{3}x_{2}-k_{4}z_{2}-k_{5}z_{1}+k_{6}$$
(58)$${\dot{z}}_{1}=\dot{u}=z_{2}$$
(59)$${\dot{z}}_{2}=-l_{1}z_{2}-\omega_{3}^{2}z_{1}+l_{3}y_{2}-l_{4}z_{1}^{3}+l_{5}z_{1}^{2}-l_{6}z_{1}+l_{7}=0$$

The fixed points of the MEMS are found and we note that the origin point is a fixed point and that the dynamic system possesses an infinite number of equilibrium points given by [28]
(60)$$E^{\cdot }(x_{1}^{\cdot },{\dot{x}}_{1}^{\cdot },y_{2}^{\cdot },{\dot{y}}_{2}^{\cdot },z_{3}^{\cdot },{\dot{z}}_{3}^{\cdot })=E(0,0,0.2\pi /\varphi_{0},0,0,0)$$

## 5. Example Calculation and Analysis

The relevant parameters used in the analysis and calculation of the examples are shown in the Table 1.

Figure 2 illustrates the amplitude–frequency diagram of the phase difference of the Josephson junction for different excitation amplitudes. It can be seen from the figure that the phase difference of the Josephson junction deviates to the right with the increase in the value of the excitation amplitudes, which means that the nonlinear term of the Josephson junction phase difference equation has a nonlinear hardening property. The solid line of the curve is the stable region of the nonlinear internal resonance, while the dash part of the curve is the unstable part. A similar representation is used for the next graph stable and unstable curves. Therefore, the phase difference solution of the Josephson junction appears to have a multi-valued phenomenon with the increase in the excitation amplitude, and there exists an unstable vibration interval in the solution. The amplitude of the solution of the phase difference of the Josephson junction equation increases with a corresponding increase in the value of the excitation amplitude.

The amplitude–frequency diagram of the circuit charge for different phase difference of the Josephson junction is shown as Figure 3. The amplitude of the circuit charge of the Josephson junction first increases and then decreases with the increase in the tuning parameters. There is no multi-valued phenomenon in the solution of the charge equation, and the solution of the equation has a stable vibration interval.

Figure 4 illustrates the amplitude frequency diagram of the amplitude variation of the nanobeam for different PDJJ when the magnetic field intensity is 3 T. It can be seen from the figure that the amplitude of the nanobeam deviates to the left with the increase in the value of amplitudes of PDJJ, which means that the nonlinear equation of the nanobeam has a nonlinear softening property. The amplitude solution of the nanobeam appears to have multi-valued phenomenon with the increase in the amplitudes of PDJJ. The amplitude of the solution of the equation increases with the increasing of amplitudes of PDJJ.

Figure 5 illustrates the amplitude frequency diagram of the variation of the amplitude of the nanobeam for different magnetic field when the excitation amplitude is 0.1. The amplitude of the nanobeam deviates to the left with the increase in the value of magnetic field. The deflection of the nanobeam under the magnetic force increases with the increase in the value of the magnetic field amplitude, and the nonlinear effect of the equation is enhanced for the couple system. The resonant vibration amplitude of the nanometer beam is sensitive to the change of magnetic field intensity, which can measure the change of magnetic field intensity. In addition, the online test method can measure the strength of the magnetic field quickly.

Figure 6 illustrates the amplitude frequency diagram of the variation of the divided resistances when the magnetic field intensity is 6 T and the excitation amplitude is 0.1. It can be seen from the figure that the amplitude of the nanobeam deviates to the left with the decrease of divided resistances. The amplitude of the nanobeam appears multi-valued phenomenon with the decrease in the value of divided resistances. Therefore, there exists an unstable vibration interval of the solution.

In this part, we will illustrate some electrodynamic behaviors of the system using phase spaces by using the fourth order Runge-Kutta integration algorithm to solve numerically Equations (54)–(59) with the zero initial conditions. Some conclusions of internal resonances for the electrodynamic system will be drawn when the excitation frequency of the generator is equal to that of the nano-beam. An analysis of the effect of the magnetic field on the system is made and some other interesting applications are listed. Figure 7 illustrates the time domain and frequency domain diagram for PDJJ for the magnetic field 6 T. The phenomenon of multi-periodic solution appears in the amplitude PDJJ, which also verifies the conclusion that the existence of nonlinear term leads to the emergence of multi-periodic solution.

Figure 8 is the time domain and frequency domain diagram for the circuit charge of different phase difference of the Josephson junction for the magnetic field 6 T. Figure 9 is the time domain and frequency domain diagram for the amplitude of the nanobeam for the magnetic field 6 T. Figure 7 and Figure 8 illustrate that the coupling and nonlinearity of the system can lead to the multiperiodic solutions. The presence of the internal resonance produces a phenomenon similar to beat vibration, indicating the flow of energy between systems. In fact, the internal resonance dynamic modal analysis is one of the goals of this study. We are particularly interested in analyzing and comparing the oscillatory properties of each type of the nanobeam coupled to the Josephson junctions and electrical resonators. Through this numerical study, we found that the unique electromagnetic characteristics of the Josephson junction have a great influence on the vibration characteristics of the nanobeam, especially the frequency and amplitude of oscillation.

Figure 10 illustrates the time domain and frequency domain diagram for amplitude of the nanobeam for the magnetic field 6 and 3 T. We evaluated the influence of the magnetic field intensity on the amplitudes, frequencies and modes of electromechanical oscillations. The amplitude of the nanobeam increases with the increase in the value of the magnetic field intensity, which is same as the conclusions as shown in Figure 5.

The involvement of the Josephson junction can estimate the effect of an external magnetic field on the coupling channel by changing the phase error in the Josephson junction. In addition, the junction current is regulated in an effective way. The coupled circuit can estimate the effect of electromagnetic induction and external magnetic field when the Josephson junction is incorporated into the circuits. The physical field effect is described by the channel current in these electronic components. The slight changes in these physical parameters are sensitive to detection in the condition of internal resonance. As a result, the physical fields can be detected by the channel current across the Josephson junction. Given the importance of the parameter of magnetic field intensity on the nanobeam vibration, we hope to bring an idea for the improvement of the amplitudes and frequencies of oscillation of the nanobeam. This idea can be also used in the field of the measurement of the magnetic field intensity.

## 6. Conclusions

The phase difference of the Josephson junction equation shows a nonlinear hardening property with the mathematical treatment method based on a Taylor series expansion. In addition, the peak amplitude of the phase difference of the Josephson junction shows an increase with the corresponding increase in the value of the excitation current. Furthermore, the circuit charge equation of the Josephson junction has a weak non-linearity, while its solution illustrates a linear behavior. Moreover, the charge solution does not show a multi-valued phenomenon, and the solution of the equation has a stable vibration interval. On the other hand, the circuit of charge increases with the corresponding increase in the value of the divider resistance. The nanobeam in the coupling of the electromechanical system has also a nonlinear softening property, and the solution of the equation appears to have a multi-valued phenomenon. Meanwhile, there is an unstable vibration interval in the solution of the nonlinear vibration of the nanobeam. Lastly, the nonlinear equation of the nanobeam dynamics becomes stronger with the increase in the value of the driving magnetic field intensity, whereas the vibration amplitude of the nanobeam decreases with the increase in the values of the voltage divider resistance. In the next research, the Josephson junction could allow the nanobeam to be used as an electro-mechanical nano actuator with a piezoelectric, photoelectric or photo-voltaic effect, and the generator can be replaced with a piezoelectric, photoelectric or photovoltaic material. This paper studies the phenomenon of internal resonances between coupled system of nanobeam and Josephson junctions. The change of magnetic field can be reflected by the change of amplitude of the nanobeam, and the magnetic field size can be measured by this principle. Therefore, the measurement of the internal amplitude can indirectly measure the magnetic field strength.

## Figures and Tables

**Figure 1 micromachines-13-01958-f001:**
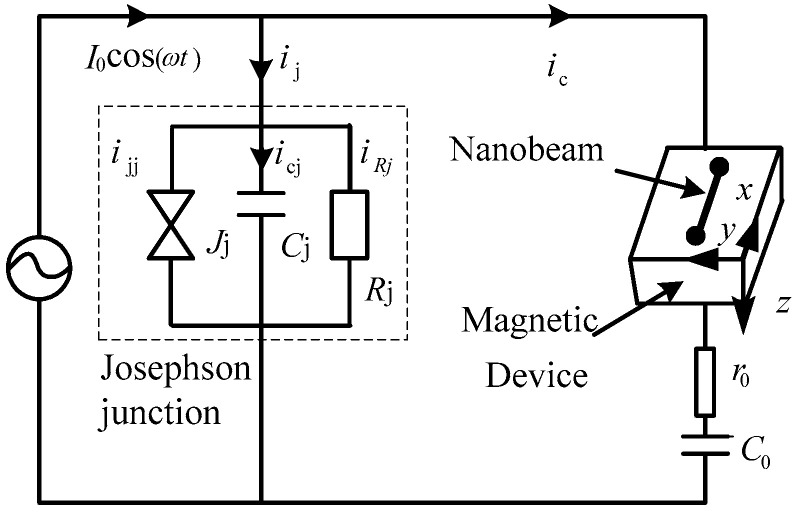
Schematic circuit of NEMS with Josephson junction.

**Figure 2 micromachines-13-01958-f002:**
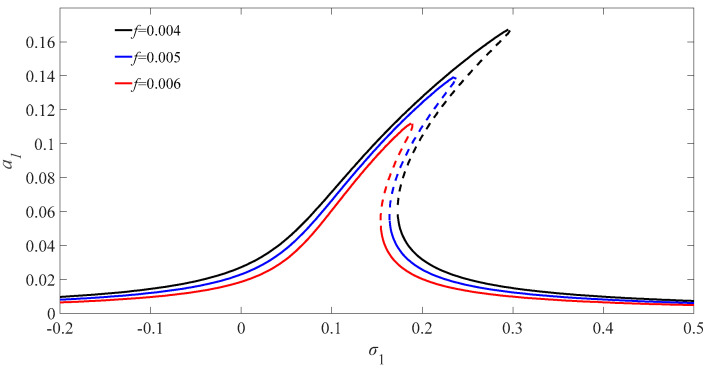
Internal resonant amplitude-frequency diagram of PDJJ for different excitation amplitudes.

**Figure 3 micromachines-13-01958-f003:**
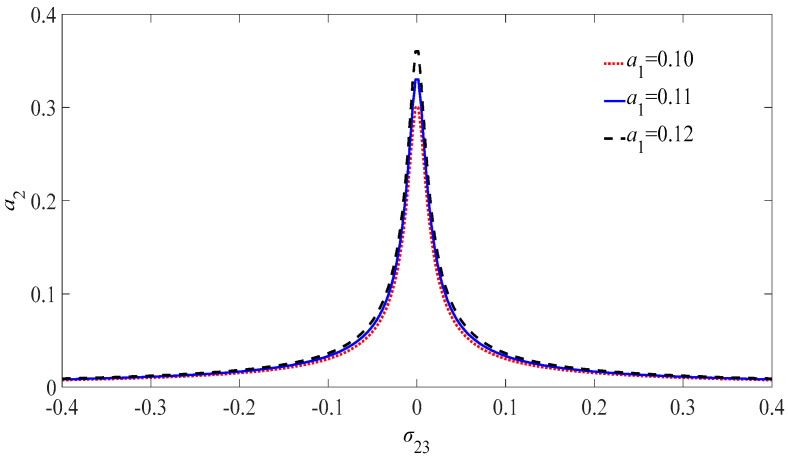
Internal resonant amplitude-frequency diagram of the circuit charge for different PDJJ.

**Figure 4 micromachines-13-01958-f004:**
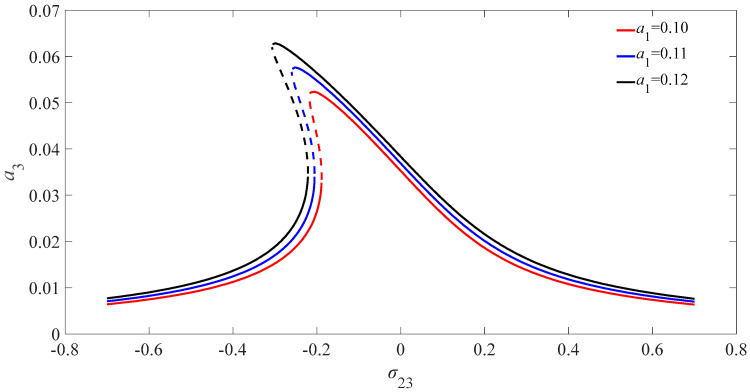
Diagram of internal resonant amplitude-frequency of nanobeam for different amplitudes of PDJJ.

**Figure 5 micromachines-13-01958-f005:**
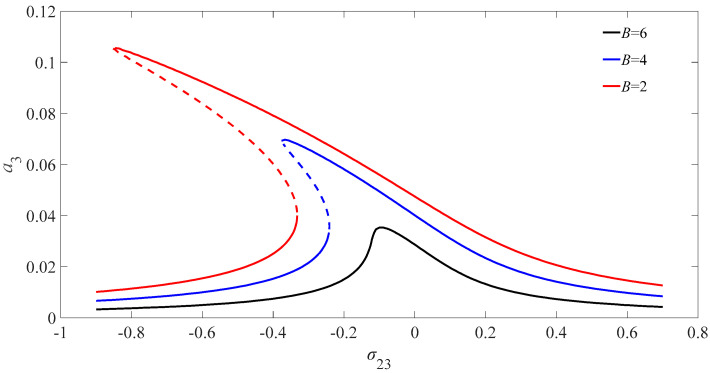
Diagram of internal resonant amplitude-frequency of nanobeam for different magnetic fields.

**Figure 6 micromachines-13-01958-f006:**
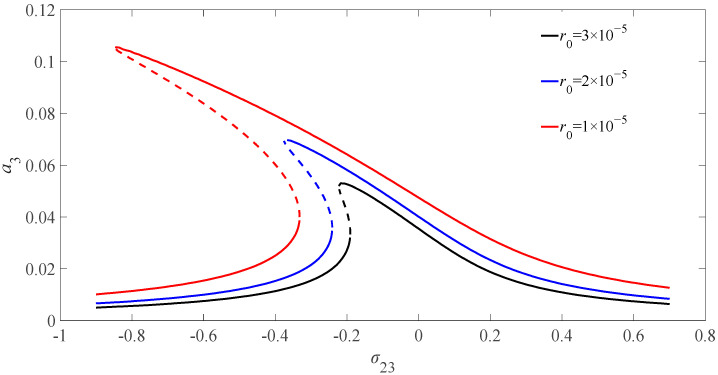
Diagram of resonant amplitude-frequency of nanobeam for different divided resistances.

**Figure 7 micromachines-13-01958-f007:**
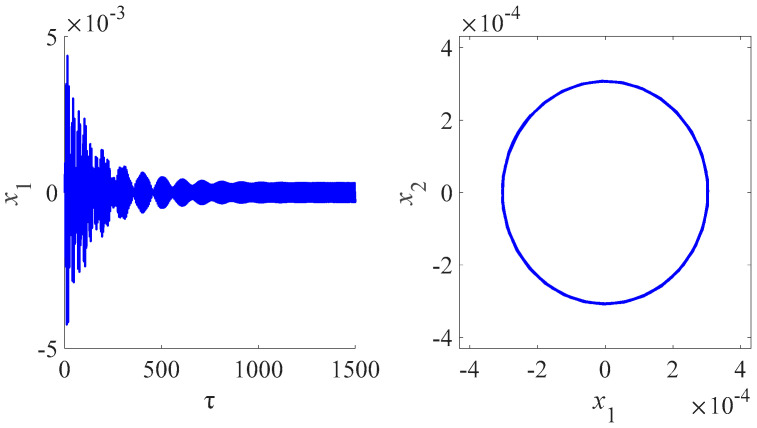
Time domain and frequency domain diagram for PDJJ for the magnetic field 6 T.

**Figure 8 micromachines-13-01958-f008:**
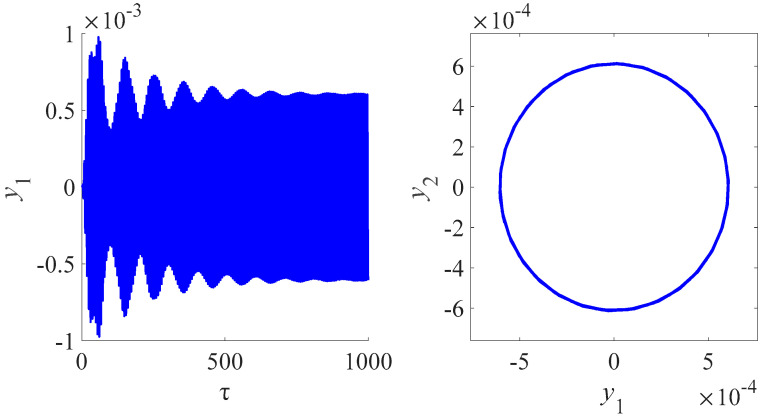
Time domain and frequency domain diagram for circuit charge of different phase difference of the Josephson junction for the magnetic field 6 T.

**Figure 9 micromachines-13-01958-f009:**
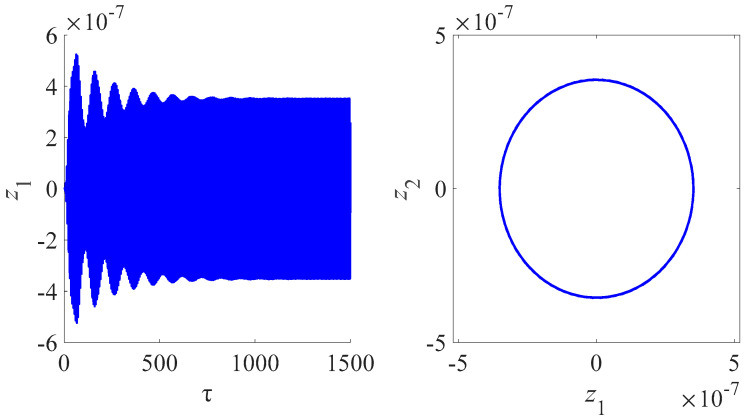
Time domain and frequency domain diagram for amplitude of the nanobeam for the magnetic field 6 T.

**Figure 10 micromachines-13-01958-f010:**
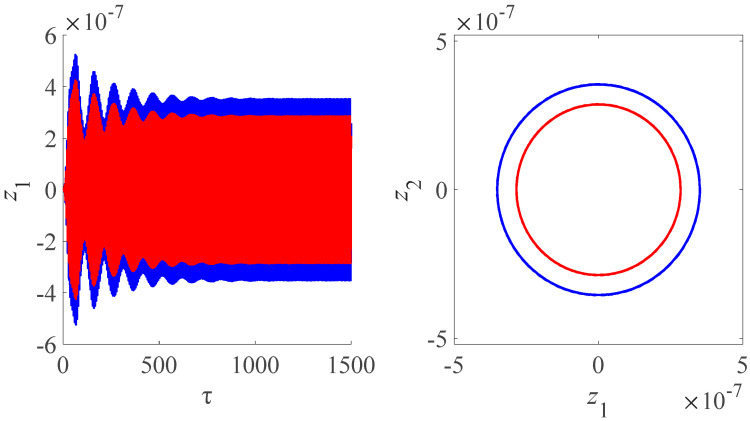
Time domain and frequency domain diagram for amplitude of the nanobeam for different. The red line is the amplitude of the nanobeam for the magnetic field 3 T. The blue line is the amplitude of the nanobeam for the magnetic field 6 T.

**Table 1 micromachines-13-01958-t001:** List of symbols and corresponding definitions.

Symbols Corresponding Definitions	Value	Units
*R*_j_ Resistance of the Josephson junction	1	Ω
*B* The magnetic field vector	6	T
*C*_j_ Self-capacitance of the Josephson junction	3 × 10^−9^	F
*C*_0_ Capacitance	9.7 × 10^−12^	F
*I*_0_ Amplitude of the excitation current	1 × 10^−10^	A
*E* Young module of the micro-beam material	50 × 10^9^	Pa
*ρ* Volume density of the micro-beam material	2330	kg/m^3^
*λ* Damping coefficient of the micro-beam	0.02	-
*r*_0_ Resistance of the divided resistor	1 × 10^−5^	Ω
*r*_p_ Electrical resistance of the micro-beam	1 × 10^−5^	Ω
*L* Length of the micro-beam	5 × 10^−7^	m
*d* wide of the micro-beam	5 × 10^−8^	m
*h* high of the micro-beam	5 × 10^−8^	m
*μ* Resistivity of the micro-beam material	0.25	
*ħ* Planck constant	6.63 × 10^−34^	
*L*_0_ Electrical inductance	1 × 10^−10^	H

## Data Availability

Raw data were generated at the computing software of large-scale facility. Derived data supporting the findings of this study are available from the corresponding author upon request.

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
