# Peer review of "Internal Resonance of the Coupling Electromechanical Systems Based on Josephson Junction Effects"

_micromachines, 2022, doi:10.3390/mi13111958_

Round 1

Author Response

Revision Report

Dear editors and reviewers,

We deeply appreciate the effort you’ve spent in reviewing our manuscript and giving advices. We have carefully revised the manuscript according to your suggestions. Our responses are listed below:

1.In Abstract, the authors demonstrated that internal resonances can induce a transfer of energy from one electromechanical system to another. However, there is no corresponding discussion in the section of “5. Example calculation and analysis”;

Answer:Due to the complexity of the energy transfer analysis process, this part is not discussed in this paper. So, we have deleted it.

2.The author can improve the Introduction by summarizing the internal resonances and mode coupling type of coupling MEMS/NEMS. The following literatures (Luo W, Gao N, Liu D. Multimode Nonlinear Coupling Induced by Internal Resonance in a Microcantilever Resonator[J]. Nano Letters, 2021, 21(2): 1062-1067; Asadi K, Yeom J, Cho H. Strong internal resonance in a nonlinear, asymmetric microbeam resonator[J]. Microsystems & nanoengineering, 2021, 7(1): 1-15; Hu K M, Zhang W M, Dong X J, et al. Scale effect on tension-induced intermodal coupling in nanomechanical resonators[J]. Journal of Vibration and Acoustics, 2015, 137(2); Asadi K, Yu J, Cho H. Nonlinear couplings and energy transfers in micro-and nano-mechanical resonators: intermodal coupling, internal resonance and synchronization[J]. Philosophical Transactions of the Royal Society A: Mathematical, Physical and Engineering Sciences, 2018, 376(2127): 20170141; Peng B, Hu K M, Fang X Y, et al. Modal characteristics of coupled MEMS resonator array under the effect of residual stress[J]. Sensors and Actuators A: Physical, 2022, 333: 113236.)

Answer:We havw cited these literatures and added related content about internal resonance and modal coupling.

3.In Introduction, the authors demonstrated that a magnetic field strength detection method is a high sensitivity and a fast on-line test method. However, the authors did not analyze and evaluate the sensitivity and response time.

Answer:We have added an explanation of this section to the description in Figure 5. The specific is " The resonance vibration amplitude of the nanometer beam is sensitive to the change of magnetic field intensity, which can measure the change of magnetic field intensity. In addition, the online test method can measure the strength of the magnetic field quickly.".

4.In Figure 1, the authors should clear show how two resonators mounted in series and coupled to a Josephson junction.

Answer:The circuit diagram of NEMS with Josephson junction has been redrawn to add a description of the Josephson Junction section.

5.There is no comparison and verification of dynamical models, especially the comparison of existing experimental data.

Answer:The nonlinear internal resonance simulation and experimental literature of Josephson Junction junction coupled nanocrystalline structures were not retrieved. In addition, the cost of the experiment is too high to be borne by the authors. This paper only starts from the simulation.

6.The writing should be improved, such as “Figure 1. Device of the NEMS with the.Josephson junctions.”; “is phase difference of The Josephson junction”; “E, ρ, μ and λ are Young module, volume density, resistivity and damping coefficient of the micro-beam material, respectivley.” ; “Ey” is not defined.

Answer:The syntax errors of some statements have been corrected.

Reviewer 2 Report

The authors investigated the dynamics of a bi-recessed microbeam coupled magnetically to two Josephson junctions. The subject is interesting. The paper can be considered for publication in Micromechanics after considering the following major points:

1. A definition should be added for all used symbols immediately after they appear in an equation. 

2- The authors should clearly define the motivation and the novelty of the paper in the introduction.

3. It is well established that nano-structures’ behavior might be size-dependent. For example, see the following refs for some of the size-dependent theories.

I) Modified Couple stress theory: “Nonlinear-electrostatic analysis of micro-actuated
beams based on couple stress and surface elasticity theories,” International Journal of
Mechanical Sciences 84, 208–217

II) Couple stress theory: Coupled effect of surface energy and size effect on the static and dynamic pull-in instability of narrow nano-switches; International Journal of Applied Mechanics 7 (04), (2015) 1550064

III) Strain gradient elasticity: Modeling the size dependent pull-in instability of beam-type NEMS using strain gradient theory; Latin American Journal of Solids and Structures 11(2014):1806-1829

Authors should at least introduce and discuss size dependency in ultra-small structures within the text. 

4. Results and discussion sections should be improved, and the authors should discuss their findings more extensively.

5. How can you validate the obtained results?

6. The following recently published papers on the impact of magnetic field on the static and dynamic performances of nano-electromechanical systems should be reviewed:

https://doi.org/10.1016/j.physe.2021.114643

 https://doi.org/10.1002/mma.7216

https://doi.org/10.1140/epjp/s13360-020-01041-z

Author Response

Revision Report

Dear editors and reviewers,

We deeply appreciate the effort you’ve spent in reviewing our manuscript and giving advices. We have carefully revised the manuscript according to your suggestions. Our responses are listed below:

1.A definition should be added for all used symbols immediately after they appear in an equation. 

Answer:The description of physical quantity symbols has been completed.

2.The authors should clearly define the motivation and the novelty of the paper in the introduction.

Answer:The novelty and motivation of writing the article have been supplemented.

3.It is well established that nano-structures’ behavior might be size-dependent. For example, see the following refs for some of the size-dependent theories.

I) Modified Couple stress theory: “Nonlinear-electrostatic analysis of micro-actuated beams based on couple stress and surface elasticity theories,” International Journal of Mechanical Sciences 84, 208–217

II) Couple stress theory: Coupled effect of surface energy and size effect on the static and dynamic pull-in instability of narrow nano-switches; International Journal of Applied Mechanics 7 (04), (2015) 1550064

III) Strain gradient elasticity: Modeling the size dependent pull-in instability of beam-type NEMS using strain gradient theory; Latin American Journal of Solids and Structures 11(2014):1806-1829

Authors should at least introduce and discuss size dependency in ultra-small structures within the text. 

Answer: The literatures on size effects have been supplemented.

4.Results and discussion sections should be improved, and the authors should discuss their findings more extensively.

Answer: Results and discussion sections have been added.

5.How can you validate the obtained results?

Answer:The nonlinear internal resonance simulation and experimental literature of Josephson Junction coupled with nano-beam structures were not retrieved. In addition, the cost of the experiment is too high to be borne by the authors. This paper only starts from the simulation.

6.The following recently published papers on the impact of magnetic field on the static and dynamic performances of nano-electromechanical systems should be reviewed:

https://doi.org/10.1016/j.physe.2021.114643

 https://doi.org/10.1002/mma.7216

https://doi.org/10.1140/epjp/s13360-020-01041-z

Answer: The related literatures on the influence of magnetic field have been added.

Reviewer 3 Report

The authors studied the internal resonances of the coupling vibration among electro-dynamic modes of a NEMS consisted by the coupling resonators connected on a Josephson junction. Some analytical and numerical studies have been done. The comments are as follows.

1 The author shows a lot of papers on internal resonance or the Josephson junction in the introduction, but lack of the literatures directly related to this paper.

2 In line 103, the author mentioned "This paper provides a magnetic field strength detection method", but I don't find this method.

3 The multi-scale method was used to solve the equation, this method depends on small parameters, but equation 14-16 has no small parameters.

4 In line 170, the author said ε is a dimensionless parameter. But actually, ε is the small parameter.

5 In line 210, "the effect of the magnetic field term can be ignored", detailed instructions or references should be given.

6 The author made a lot of simplifications, for example, in line 218 phi3=phi4, the term with A3 in equation 44 was ignored, and in line 227 sigma2=sigma3, but they did not explain why. Whether the simplifications have a great impact on the results had not been verified.

7 Equations 54-59 are clearly a non-autonomous system, not the autonomous system mentioned in line 246. And there is something wrong with equation 59.

8 What is the y2=2pi/phi0 in the equilibrium point of equation 60?

9 In line 265, the author mentions that figure 2 is softening, in fact, figure 2 is obvious hardening, and in line 284, figure 4 is softening, but the author thinks it is hardening.

10 In figures 2, 4, 5 and 6, the author should clearly and quantitatively show the stable and unstable regions.

11 According to the author's expression, the data corresponding to the black line in figure 4 and the red dot line in figure 5 should be the same, but the results are obviously different, and I don't know why.

12 According to the author's data, r0 in figure 5 should be 1e-4, but it is clear that the black line in figure 5 and r0=3e-5 in figure 6 are the same.

13 Figure 7-9 shows the Time domain and frequency domain diagram, but I don't find the results in the frequency domain. Actually, the author gives the time process and phase trajectory.

14 All the analytical results in this paper are lack of numerical verification. All the figures shown in the paper do not give enough data. The results are too simple to support the author's conclusion.

Author Response

Revision Report

Dear editors and reviewers,

We deeply appreciate the effort you’ve spent in reviewing our manuscript and giving advices. We have carefully revised the manuscript according to your suggestions. Our responses are listed below:

1.The author shows a lot of papers on internal resonance or the Josephson junction in the introduction, but lack of the literatures directly related to this paper.

Answer: There are few literatures related to this article. The authors have not found similar literatures.

2 In line 103, the author mentioned "This paper provides a magnetic field strength detection method", but I don't find this method.

Answer: This paper studies the phenomenon of internal resonances between coupled system of nanobeam and Josephson junctions. The magnetic field change can be reflected by the change of amplitude, and the magnetic field size can be measured by this principle. Therefore, the measurement of the internal common amplitude can indirectly measure the magnetic field strength.

3 The multi-scale method was used to solve the equation, this method depends on small parameters, but equation 14-16 has no small parameters.

Answer: The author has revised it.

4 In line 170, the author said ε is a dimensionless parameter. But actually, ε is the small parameter.

Answer: The author has revised it.

5 In line 210, "the effect of the magnetic field term can be ignored", detailed instructions or references should be given.

Answer: Since the phase difference is an Angle difference, its value is much larger than the vibration amplitude of the nanobeam. The details can see Fig. 1 for comparison with Fig. 3, Fig.7 and Fig.9.

6 The author made a lot of simplifications, for example, in line 218 phi3=phi4, the term with A3 in equation 44 was ignored, and in line 227 sigma2=sigma3, but they did not explain why. Whether the simplifications have a great impact on the results had not been verified.

Answer: These variables include the tuning parameters, which are the frequency parameters that regulate the internal resonance. Under the condition of internal resonance, the approximate equality condition can be satisfied by changing the tuning parameter value.

7 Equations 54-59 are clearly a non-autonomous system, not the autonomous system mentioned in line 246. And there is something wrong with equation 59.

Answer:This is a misstatement, it should be non-autonomous system.

8 What is the y2=2pi/phi0 in the equilibrium point of equation 60?

Answer:Numerical values of equilibrium points refer to [28].

9 In line 265, the author mentions that figure 2 is softening, in fact, figure 2 is obvious hardening, and in line 284, figure 4 is softening, but the author thinks it is hardening.

Answer:There was an error in the original statement, which has been modified.

10 In figures 2, 4, 5 and 6, the author should clearly and quantitatively show the stable and unstable regions.

Answer:The stability and instability curves have been re-marked.

11 According to the author's expression, the data corresponding to the black line in figure 4 and the red dot line in figure 5 should be the same, but the results are obviously different, and I don't know why.

Answer: There was an error in the description of the magnetic field strength. It should be 3T instead of 2T.

12 According to the author's data, r0 in figure 5 should be 10-4, but it is clear that the black line in figure 5 and r0=3.0*10-5 in figure 6 are the same.

Answer: There is an error in the numerical description of the resistance. It should be 10-5 Ω instead of the original 10-4 Ω. Figure 6 has been modified.

13 Figure 7-9 shows the Time domain and frequency domain diagram, but I don't find the results in the frequency domain. Actually, the author gives the time process and phase trajectory.

 Answer: These three images compare the response to the magnetic field sensitivity mainly from the amplitude. In addition, since the journal only allowed 7 days for revision, the frequency domain map cannot be supplemented now.

14 All the analytical results in this paper are lack of numerical verification. All the figures shown in the paper do not give enough data. The results are too simple to support the author's conclusion.

Answer: Due to the high coupling degree of the system equations, the authors have not found an appropriate method to conduct in-depth numerical analysis and comparison verification. This paper only analyses the approximate solution and the numerical solution. The authors improved and supplemented the image data in the paper. The image captions have been further supplemented.

Round 2

Reviewer 2 Report

The authors improved the paper. It can be considered for publication in the present form.

Author Response

Dear editors and reviewers,

We deeply appreciate the effort you’ve spent in reviewing our manuscript and giving advices. We have carefully revised the manuscript according to your suggestions. Our responses are listed below:

1.The solid line of the curve is the stable region of the nonlinear internal resonance, while the dash part of the curve is the unstable part in figures 2,4,5 and 6.

2. Further corrections have been made to the grammatical errors in the whole paper.

Reviewer 3 Report

Dear editor,

The author has made a good revision to the manuscript. I think the paper can be accepted after minor revision.

Overall Recommendation

( ) Accept in present form
(√) Accept after minor revision 
(corrections to minor methodological errors and text editing)
( ) Reconsider after major revision 
(control missing in some experiments)
( ) Reject 
(article has serious flaws, additional experiments needed, research not conducted correctly)

Comments and Suggestions for Authors

The stable and unstable regions of the amplitude-frequency curve should be accurately expressed. For example, stable solutions are represented by solid lines, and unstable ones are represented by dash lines. The format of the figures is given in the attchment.

Author Response

Dear editors and reviewers,

We deeply appreciate the effort you’ve spent in reviewing our manuscript and giving advices. We have carefully revised the manuscript according to your suggestions. Our responses are listed below:

1.The solid line of the curve is the stable region of the nonlinear internal resonance, while the dash part of the curve is the unstable part in figures 2,4,5 and 6.

2.Further corrections have been made to the grammatical errors in the whole paper.
